# PerspectiveNet: 3D Object Detection from a Single RGB Image via Perspective Points

**Siyuan Huang**
Department of Statistics
huangsiyuan@ucla.edu

**Yixin Chen**
Department of Statistics
ethanchen@ucla.edu

**Tao Yuan**
Department of Statistics
taoyuan@ucla.edu

**Siyuan Qi**
Department of Computer Science
syqi@cs.ucla.edu

**Yixin Zhu**
Department of Statistics
yixin.zhu@ucla.edu

**Song-Chun Zhu**
Department of Statistics
sczhu@stat.ucla.edu

## Abstract

Detecting 3D objects from a single RGB image is intrinsically ambiguous, thus requiring appropriate prior knowledge and intermediate representations as constraints to reduce the uncertainties and improve the consistencies between the 2D image plane and the 3D world coordinate. To address this challenge, we propose to adopt perspective points as a new intermediate representation for 3D object detection, defined as the 2D projections of local Manhattan 3D keypoints to locate an object; these perspective points satisfy geometric constraints imposed by the perspective projection. We further devise PerspectiveNet, an end-to-end trainable model that simultaneously detects the 2D bounding box, 2D perspective points, and 3D object bounding box for each object from a single RGB image. PerspectiveNet yields three unique advantages: (i) 3D object bounding boxes are estimated based on perspective points, bridging the gap between 2D and 3D bounding boxes *without* the need of category-specific 3D shape priors. (ii) It predicts the perspective points by a *template-based* method, and a perspective loss is formulated to maintain the perspective constraints. (iii) It maintains the consistency between the 2D perspective points and 3D bounding boxes via a *differentiable* projective function. Experiments on SUN RGB-D dataset show that the proposed method significantly outperforms existing RGB-based approaches for 3D object detection.

## 1 Introduction

> If one hopes to achieve a full understanding of a system as complicated as a nervous system, ..., or even a large computer program, then one must be prepared to contemplate different kinds of explanation at different levels of description that are linked, at least in principle, into a cohesive whole, even if linking the levels in complete details is impractical. — David Marr [1], pp. 20–21

In a classic view of computer vision, David Marr [1] conjectured that the perception of a 2D image is an *explicit* multi-phase information process, involving (i) an early vision system of perceiving textures [2, 3] and textons [4, 5] to form a primal sketch as a perceptually lossless conversion from the raw image [6, 7], (ii) a mid-level vision system to construct 2.1D (multiple layers with partial occlusion) [8–10] and 2.5D [11] sketches, and (iii) a high-level vision system that recovers the full 3D [12–14]. In particular, he highlighted the importance of different levels of organization and the internal representation [15].

In parallel, the school of Gestalt Laws [16–23] and perceptual organization [24, 25] aims to resolve the 3D reconstruction problem from a single RGB image without forming the depth cues; but rather, they often use some sorts of priors—groupings and structural cues [26, 27] that are likely to be invariant over wide ranges of viewpoints [28], resulting in the birth of the SIFT feature [29]. Later, from a Bayesian perspective at a scene level, such priors, independent of any 3D scene structures,

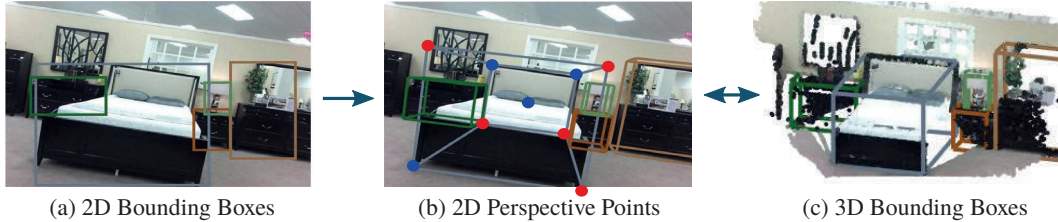

| (a) 2D Bounding Boxes | (b) 2D Perspective Points | (c) 3D Bounding Boxes |

Figure 1: Traditional 3D object detection methods directly estimate (c) the 3D object bounding boxes from (a) the 2D bounding boxes, which suffer from the uncertainties between the 2D image plane and the 3D world. The proposed PerspectiveNet utilizes (b) the 2D perspective points as the intermediate representation to bridge the gap. The perspective points are the 2D perspective projection of the 3D bounding box corners, containing rich 3D information (*e.g.*, positions, orientations). The red dots indicate the perspective points of the bed that are challenging to emerge based on the visual features, but could be inferred by the context (correlations and topology) among other perspective points.

were found in the human-made scenes, known as the Manhattan World assumption [30]. Importantly, further studies found that such priors help to improve object detection [31].

In this paper, inspired by these two classic schools in computer vision, we seek to test the following two hypotheses using modern computer vision methods: (i) Could an *intermediate representation* facilitate modern computer vision tasks? (ii) Is such an intermediate representation a better and more *invariant prior* compared to the priors obtained directly from specific tasks?

In particular, we tackle the challenging task of 3D object detection from a single RGB image. Despite the recent success in 2D scene understanding (*e.g.*, [32, 33], there is still a significant performance gap for 3D computer vision tasks based on a single 2D image. Recent modern approaches directly regress the 3D bounding boxes [34–36] or reconstruct the 3D objects with specific 3D object priors [37–40]. In contrast, we propose an end-to-end trainable framework, PerspectiveNet, that sequentially estimates the 2D bounding box, 2D perspective points, and 3D bounding box for each object with a local Manhattan assumption [41], in which the perspective points serve as the intermediate representation, defined as the 2D projections of local Manhattan 3D keypoints to locate an object.

The proposed method offers three unique advantages. First, the use of perspective points as the *intermediate representation* bridges the gap between 2D and 3D bounding boxes *without* utilizing any extra category-specific 3D shape priors. As shown in Figure 1, it is often challenging for learning-based methods to estimate the 3D bounding boxes from 2D images directly; regressing 3D bounding boxes from 2D input is a highly under-constrained problem and can be easily influenced by appearance variations of shape, texture, lighting, and background. To alleviate this issue, we adopt the perspective points as an intermediate representation to represent the local Manhattan frame that each 3D object aligns with. Intuitively, the perspective points of an object are *3D geometric constraints in the 2D space*. More specifically, the 2D perspective points for each object are defined as the perspective projection of the 3D object bounding box (concatenated with its center), and each 3D box aligns within a 3D local Manhattan frame. These perspective points are fused into the 3D branch to predict the 3D attributes of the 3D bounding boxes.

Second, we devise a *template-based* method to efficiently and robustly estimate the perspective points. Existing methods [42–44, 33, 45] usually exploit heatmap or probability distribution map as the representation to learn the location of visual points (*e.g.*, object keypoint, human skeleton, room layout), relying heavily on the view-dependent visual features, thus insufficient to resolve occlusions or large rotation/viewpoint changes in complex scenes; see an example in Figure 1 (b) where the five perspective points (in red) are challenging to emerge from pure visual features but could be inferred by the correlations and topology among other perspective points. To tackle this problem, we treat each set of 2D perspective points as the low dimensional embedding of its corresponding set of 3D points with a constant topology; such an embedding is learned by predicting the perspective points as a mixture of sparse templates. A perspective loss is formulated to impose the perspective constraints; the details are described in § 3.2.

Third, the consistency between the 2D perspective points and 3D bounding boxes can be maintained by a *differentiable* projective function; it is end-to-end trainable, from the 2D region proposals, to the 2D bounding boxes, to the 2D perspective points, and to the 3D bounding boxes.

In the experiment, we show that the proposed PerspectiveNet outperforms previous methods with a large margin on SUN RGB-D dataset [46], demonstrating its efficacy on 3D object detection.

## 2 Related Work

**3D object detection from a single image**    Detecting 3D objects from a single RGB image is a challenging problem, particularly due to the intrinsic ambiguity of the problem. Existing methods could be categorized into three streams: (i) geometry-based methods that estimate the 3D bounding boxes with geometry and 3D world priors [47–51]; (ii) learning-based methods that incorporate category-specific 3D shape prior [52, 38, 40] or extra 2.5D information (depth, surface normal, and segmentation) [37, 39, 53] to detect 3D bounding boxes or reconstruct the 3D object shape; and (iii) deep learning methods that directly estimates the 3D object bounding boxes from 2D bounding boxes [54, 34–36]. To make better estimations, various techniques have been devised to enforce consistencies between the estimated 3D and the input 2D image. Huang et al. [36] proposed a two-stage method to learn the 3D objects and 3D layout cooperatively. Kundu et al. [37] proposed a 3D object detection and reconstruction method using category-specific object shape prior by render-and-compare. Different from these methods, the proposed PerspectiveNet is a one-stage end-to-end trainable 3D object detection framework using perspective points as an intermediate representation; the perspective points naturally bridge the gap between the 2D and 3D bounding boxes without any extra annotations, category-specific 3D shape priors, or 2.5D maps.

**Manhattan World assumption**    Human-made environment, from the layout of a city to structures such as buildings, room, furniture, and many other objects, could be viewed as a set of parallel and orthogonal planes, known as the Manhattan World (MW) assumption [31]. Formally, it indicates that most human-made structures could be approximated by planar surfaces that are parallel to one of the three principal planes of a common orthogonal coordinate system. This strict Manhattan World assumption is later extended by a Mixture of Manhattan Frame (MMF) [55] to represent more complex real-world scenes (*e.g.*, city layouts, rotated objects). In literature, MW and MMF have been adopted in vanish points (VPs) estimation and camera calibration [56, 57], orientation estimation [58–60], layout estimation [61–64, 44], and 3D scene reconstruction [65–67, 41, 68, 69]. In this work, we extend the MW to local Manhattan assumption where the cuboids are aligned with the vertical (gravity) direction but with arbitrary horizontal orientation (also see Xiao and Furukawa [41]), and perspective points are adopted as the intermediate representation for 3D object detection.

**Intermediate 3D representation**    Intermediate 3D representations are bridges that narrow the gap and maintain the consistency between the 2D image plane and 3D world. Among them, 2.5D sketches have been broadly used in reconstructing the 3D shapes [70–72] and 3D scenes [73, 38]. Other recent alternative intermediate 3D representations include: (i) Wu et al. [74] uses pre-annotated and category-specific object keypoints as an intermediate representation, and (ii) Tekin et al. [75] uses projected corners of 3D bounding boxes in learning the 6D object pose. In this paper, we explore the perspective points as an intermediate representation of 2D and 3D bounding boxes, and provide an efficient learning framework for 3D object detection.

## 3 Learning Perspective Points for 3D Object Detection

### 3.1 Overall Architecture

As shown in Figure 2, the proposed PerspectiveNet contains a backbone architecture for feature extraction over the entire image, a region proposal network (RPN) [32] that proposes regions of interest (RoIs), and a network head including three region-wise parallel branches. For each proposed box, its RoI feature is fed into the three network branches to predict: (i) the object class and the 2D bounding box offset, (ii) the 2D perspective points (projected 3D box corners and object center) as a weighted sum of predicted perspective templates, and (iii) the 3D box size, orientation, and its distance from the camera. Detected 3D boxes are reconstructed by the projected object center, distance, box size, and rotation. The overall architecture of the PerspectiveNet resembles the R-CNN structure, and we refer readers to [32, 76, 33] for more details of training R-CNN detectors.

During training, we define a multi-task loss on each proposed RoI as

$$\mathcal{L} = \mathcal{L}_{cls} + \mathcal{L}_{2D} + \mathcal{L}_{pp} + \mathcal{L}_p + \mathcal{L}_{3D} + \mathcal{L}_{proj}, \tag{1}$$

where the classification loss $\mathcal{L}_{cls}$ and 2D bounding box loss $\mathcal{L}_{2D}$ belong to the 2D bounding box branch and are identical to those defined in 2D R-CNNs [32, 33]. $\mathcal{L}_{pp}$ and $\mathcal{L}_p$ are defined on the perspective point branch (§ 3.2), $\mathcal{L}_{3D}$ is defined on the 3D bounding box branch (see § 3.3), and the $\mathcal{L}_{proj}$ is defined on maintaining the 2D-3D projection consistency (see § 3.4).

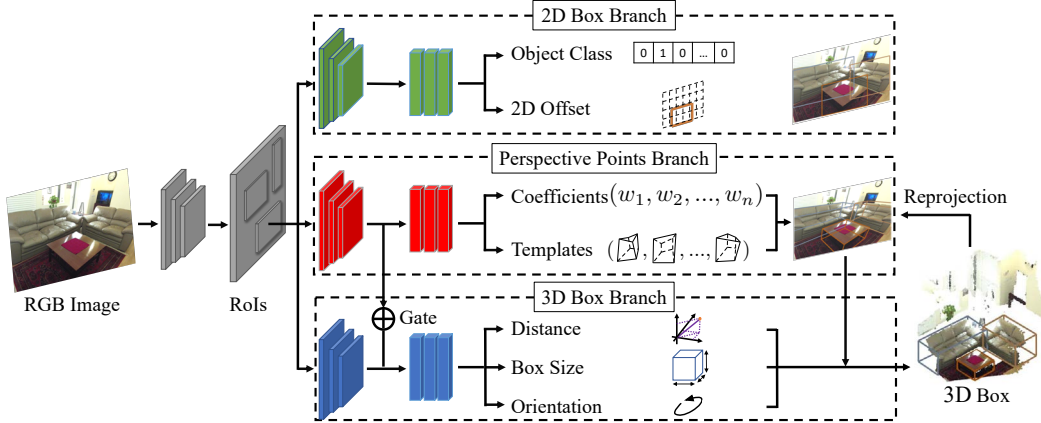

Figure 2: The proposed framework of the PerspectiveNet. Given an RGB Image, the backbone of PerspectiveNet extracts global features and propose candidate 2D bounding boxes (RoIs). For each proposed box, its RoI feature is fed into three network branches to predict: (i) the object class and the 2D box offset, (ii) 2D perspective templates (projected 3D box corners and object center) and the corresponding coefficients, and (iii) the 3D box size, orientation, and its distance from the camera. Detected 3D boxes are reconstructed by the projected object center, distance, box size, and rotation. By projecting the detected 3D boxes to 2D and comparing them with 2D perspective points, the network imposes and learns a consistency between the 2D inputs and 3D estimations.

## 3.2 Perspective Point Estimation

The perspective point branch estimates the set of 2D perspective points for each RoI. Formally, the 2D perspective points of an object are the 2D projections of local Manhattan 3D keypoints to locate that object, and they satisfy certain geometric constraints imposed by the perspective projection. In our case, the perspective points (Figure 1(b)) include the 2D projections of the 3D bounding box corners and the 3D object center. The perspective points are predicted using a template-based regression and learned by a mean squared error and a perspective loss detailed below.

### 3.2.1 Template-based Regression

Most of the existing methods [42–44, 33, 45] estimate visual keypoints with heatmaps, where each map predicts the location for a certain keypoint. However, predicting perspective points by heatmaps has two major problems: (i) Heatmap prediction for different keypoints is independent, thus fail to capture the topology correlations among the perspective points. (ii) Heatmap prediction for each keypoint relies heavily on the visual feature such as corners, which may be difficult to detect (see an example in Figure 1(b)). In contrast, each set of 2D perspective points can be treated as a low dimensional embedding of a set of 3D points with a particular topology, thus inferring such points relies more on the relation and topology among the points instead of just the visual features.

To tackle these problems, we avoid dense per-pixel predictions. Instead, we estimate the perspective points by a mixture of sparse templates [77, 78]. The sparse templates are more robust when facing unfamiliar scenes or objects. Ablative experiments show that the proposed template-based method provides a more accurate estimation of perspective points than heatmap-based methods; see § 5.1.

Specifically, we project both the 3D object center and eight 3D bounding box corners to 2D with camera parameters to generate the ground-truth 2D perspective points $P_{gt} \in \mathbb{R}^{2 \times 9}$. Since a portion of the perspective points usually lies out of the RoI, we calculate the location of the perspective points in an extended (doubled) size of RoI and normalize the locations to $[0, 1]$.

We predict the perspective points by a linear combination of templates; see Figure 3. The perspective point branch has a $C \times K \times 2 \times 9$ dimensional output for the templates $T$, and a $C \times K$ dimensional output for the coefficients $w$, where $K$ denotes the number of templates for each class and $C$ denotes the number of object classes. The templates $T$ is scaled to $[0, 1]$ by a sigmoid nonlinear function, and the coefficients $w$ is normalized by a softmax function. The estimated perspective points $\hat{P} \in \mathbb{R}^{C \times 2 \times 9}$ can be computed by a linear combination:

$$\hat{P}_i = \sum_{k=1}^{K} w_{ik} T_{ik}, \quad \forall i = 1, \cdots, C. \tag{2}$$

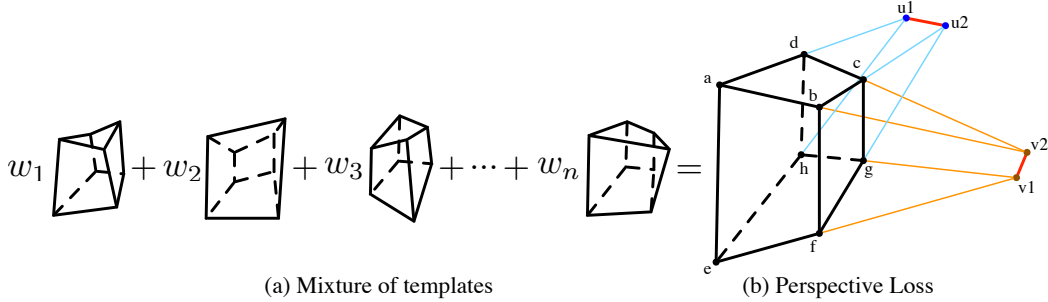

<div align="center">(a) Mixture of templates          (b) Perspective Loss</div>

Figure 3: Perspective point estimation. (a) The perspective points are estimated by a mixture of templates through a linear combination. Each template encodes geometric cues including orientations and viewpoints. (b) The perspective loss enforces each set of 2D perspective points to be the perspective projection of a (vertical) 3D cuboid. For a vertical cuboid, the projected vertical edges (*i.e.*, $ae$, $bf$, $cg$, and $dh$) should be parallel or near parallel (under small camera tilting angles). For 3D parallel lines that are perpendicular to the gravity direction, the vanishing points of their 2D projections should coincide (*e.g.*, $u1$ and $u2$).

The template design is both class-specific and instance-specific: (i) Class-specific: we decouple the prediction of the perspective point and the object class, allowing the network to learn perspective points for every class without competition among classes. (ii) Instance-specific: the templates are inferred for each RoI; hence, they are specific to each object instance. The templates are automatically learned for each object instance from data with the end-to-end learning framework; thus, both the templates and coefficients for each instance are optimizable and can better fit the training data.

The average mean squared error (MSE) loss is defined as $\mathcal{L}_{pp} = \mathrm{MSE}(\hat{P}_c, P_{gt})$. For an RoI associated with ground-truth class $c$, $\mathcal{L}_{pp}$ is only defined on the $c$'s perspective points during training; perspective point outputs from other classes do not contribute to the loss. In inference, we rely on the dedicated classification branch to predict the class label to select the output perspective points.

### 3.2.2 Perspective Loss

Under the assumption that each 3D bounding box aligns with a local Manhattan frame, we regularize the estimation of the perspective points to satisfy the constraint of perspective projection. Each set of mutually parallel lines in 3D can be projected into 2D as intersecting lines; see Figure 3 (b). These intersecting lines should converge at the same vanishing point. Therefore, the desired algorithm would penalize the distance between the intersection points from the two sets of intersecting lines. For example in Figure 3 (b), we select line $ad$ and line $eh$ as a pair of lines, $bc$ and $fg$ as another, and compute the distance between their intersection point $u_1$ and $u_2$. Additionally, since we assume each 3D local Manhattan frame aligns with the vertical (gravity) direction, we enforce the edges in gravity direction (*i.e.*, $ae$, $bf$, $cg$, and $dh$) to be parallel by penalizing the large slope variance.

The perspective loss is computed as $\mathcal{L}_p = \mathcal{L}_{d1} + \mathcal{L}_{d2} + \mathcal{L}_{grav}$, where $\mathcal{L}_{grav}$ penalizes the slope variance in gravity direction, $\mathcal{L}_{d1}$ and $\mathcal{L}_{d2}$ penalize the intersection point distance for the two perpendicular directions along the gravity direction.

### 3.3 3D Bounding Box Estimation

Estimating 3D bounding boxes is a two-step process. In the first step, the 3D branch estimates the 3D attributes, including the distance between the camera center and the 3D object center, as well as the 3D size and orientation following Huang et al. [36]. Since the perspective point branch encodes rich 3D geometric features, the 3D attribute estimator aggregates the feature from perspective point branch with a soft gated function between $[0, 1]$ to improve the prediction. The gated function serves as a soft-attention mechanism that decides how much information from perspective points should contribute to the 3D prediction.

In the second step, with the estimated projected 3D bounding boxes center (*i.e.*, the first estimated perspective point) and the 3D attributes, we compose the 3D bounding boxes by the inverse projection from the 2D image plane to the 3D world following Huang et al. [36] given camera parameters.

The 3D loss is computed by the sum of individual losses of 3D attributes and a joint loss of 3D bounding box $\mathcal{L}_{3D} = \mathcal{L}_{dis} + \mathcal{L}_{size} + \mathcal{L}_{ori} + \mathcal{L}_{box3d}$.

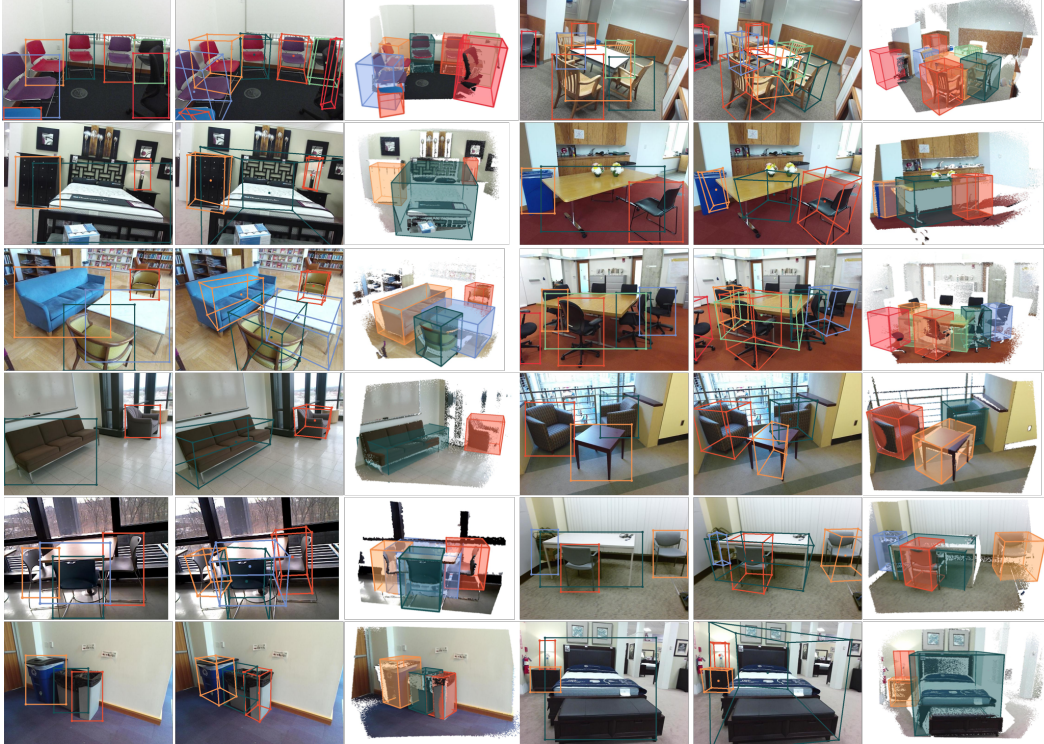

Figure 4: Qualitative results (top 50%). For every three columns as a group: (Left) The RGB image with 2D detection results. (Middle) The RGB image with estimated perspective points. (Right) The results in 3D point cloud; point cloud is used for visualization only.

### 3.4 2D-3D Consistency

In contrast to prior work [74, 79, 80, 35, 70, 36] that enforces the consistency between estimated 3D objects and 2D image, we devise a new way to impose a re-projection consistency loss between 3D bounding boxes and perspective points. Specifically, we compute the 2D projected perspective points $P_{proj}$ by projecting the 3D bounding box corners back to 2D image plane and computing the distance with respect to ground-truth perspective points $\mathcal{L}_{proj} = \text{MSE}(P_{proj}, P_{gt})$. Comparing with prior work to maintain the consistency between 2D and 3D bounding boxes by approximating the 2D projection of 3D bounding boxes [35, 36], the proposed method uses the *exact* projection of projected 3D boxes to establish the consistency, capturing a more precise 2D-3D relationship.

## 4 Implementation Details

**Network Backbone** Inspired by He et al. [33], we use the combination of residual network (ResNet) [81] and feature pyramid network (FPN) [82] to extract the feature from the entire image. A region proposal network (RPN) [32] is used to produce object proposals (*i.e.*, RoI). A RoIAlign [33] module is adopted to extract a smaller features map ($256 \times 7 \times 7$) for each proposal.

**Network Head** The network head consists of three branches, and each branch has its individual feature extractor and predictor. Three feature extractors have the same architecture of two fully connected (FC) layers; each FC layer is followed by a ReLU function. The feature extractors take the $256 \times 7 \times 7$ dimensional RoI features as the input and output a 1024 dimensional vector.

The predictor in the 2D branch has two separate FC layers to predict a $C$ dimensional object class probabilities and a $C \times 4$ dimensional 2D bounding box offset. The predictor in the perspective point branch predicts $C \times K \times 2 \times 9$ dimensional templates and $C \times K$ dimensional coefficients with two FC layers and their corresponding nonlinear activation functions (*i.e.*, sigmoid, softmax). The soft gate in the 3D branch consists of an FC layer (1024-1) and a sigmoid function to generate the weight for feature aggregation. The predictor in the 3D branch consists of three FC layers to predict the size, the distance from the camera, and the orientation of the 3D bounding box.

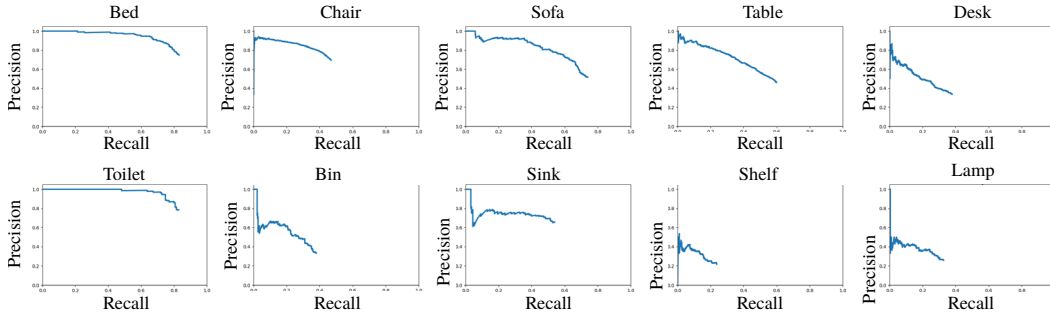

Figure 5: Precision-Recall (PR) curves for 3D object detection on SUN RGB-D

# 5 Experiments

**Dataset** We conduct comprehensive experiments on SUN RGB-D [46] dataset. The SUN RGB-D dataset has a total of 10,335 images, in which 5,050 are test images. It has a rich annotation of scene categories, camera pose, and 3D bounding boxes. We evaluate the 3D object detection results of the proposed PerspectiveNet, make comparisons with the state-of-the-art methods, and further examine the contribution of each module in ablative experiments.

**Experimental Setup** To prepare valid data for training the proposed model, we discard the images with no 3D objects or incorrect correspondence between 2D and 3D bounding boxes, resulting 4783 training images and 4220 test images. We detect 30 categories of objects following Huang et al. [36].

**Reproduciblity Details** During training, an RoI is considered positive if it has the IoU with a ground-truth box of at least 0.5. $\mathcal{L}_{pp}$, $\mathcal{L}_p$, $\mathcal{L}_{3D}$, and $\mathcal{L}_{proj}$ are only defined on positive RoIs. Each image has N sampled RoIs, where the ratio of positive to negative is 1:3 following the protocol presented in Girshick [76].

We resize the images so that the shorter edges are all 800 pixels. To avoid over-fitting, a data augmentation procedure is performed by randomly flipping the images or randomly shifting the 2D bounding boxes with corresponding labels during the training. We use SGD for optimization with a batch size of 32 on a desktop with 4 Nvidia TITAN RTX cards (8 images each card). The learning rate starts at 0.01 and decays by 0.1 at 30,000 and 35,000 iterations. We implement our framework based on the code of Massa and Girshick [83]. It takes 6 hours to train, and the trained PerspectiveNet provides inference in real-time (20 FPS) using a single GPU.

Since the consistency loss and perspective loss can be substantial during the early stage of the training process, we add them to the joint loss when the learning rate decays twice. The hyper-parameter (*e.g.*, the weights of losses, the architecture of network head) is tuned empirically by a local search.

**Evaluation Metric** We evaluate the performance of 3D object detection using the metric presented in Song et al. [46]. Specifically, we first calculate the 3D Intersection over Union (IoU) between the predicted 3D bounding boxes and the ground-truth 3D bounding boxes, and then compute the mean average precision (mAP). Following Huang et al. [36], we set the 3D IoU threshold as 0.15 in the absence of depth information.

**Qualitative Results** The qualitative results of 2D object detection, 2D perspective point estimation, and 3D object detection are shown in Figure 4. Note that the proposed method performs accurate 3D object detection in some challenging scenes. For the perspective point estimation, even though some of the perspective points are not aligned with image features, the proposed method can still localize their positions robustly.

**Quantitative Results** Since the state-of-the-art method [36] learns the camera extrinsic parameters jointly, we provide two protocals for evaluations for a fair comparison: (i) PerspectiveNet given ground-truth camera extrinsic parameter (*full*), and (ii) PerspectiveNet without ground-truth camera extrinsic parameter by learning it jointly following [36] (*w/o. cam*).

We learn the detector for 30 object categories and report the precision-recall (PR) curve of 10 main categories in Figure 5. We calculate the area under the curve to compute AP; Table 1 shows the comparisons of APs of the proposed models with existing approaches (see supplementary materials for the APs of all 30 categories).

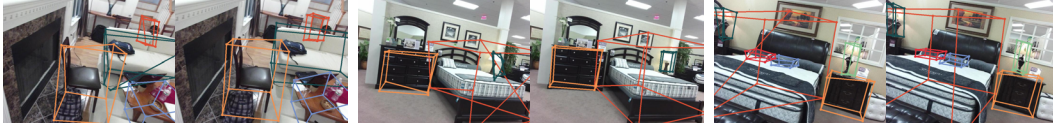

Figure 6: Heatmaps vs. templates for perspective point prediction. (Left) Estimated by heatmap-based method. (Right) Estimated by the proposed template-based method.

Note that the critical difference between the proposed model and the state-of-the-art method [36] is the intermediate representation to learn the 2D-3D consistency. Huang et al. [36] uses 2D bounding boxes to enforce a 2D-3D consistency by minimizing the differences between projected 3D boxes and detected 2D boxes. In contrast, the proposed intermediate representation has a clear advantage since projected 3D boxes often are not 2D rectangles, and perspective points eliminate such errors.

Quantitatively, our full model improves the mAP of the state-of-the-art method [36] by 14.71%, and the model without the camera extrinsic parameter improves by 10.91%. The significant improvement of the mAP demonstrates the efficacy of the proposed intermediate representation. We defer more analysis on how each component contributes to the overall performance in § 5.1.

## 5.1 Ablative Analysis

In this section, we analyze each major component of the model to examine its contribution to the overall significant performance gain. Specifically, we design six variants of the proposed model.
- $S_1$: The model trained without the perspective point branch, using the 2D offset to predict the 3D center of the object following Huang et al. [36].
- $S_2$: The model that aggregates the feature from the perspective point branch and 3D branch directly without the gate function.
- $S_3$: The model that aggregates the feature from the perspective point branch and 3D branch with a gate function that only outputs 0 or 1 (hard gate).
- $S_4$: The model trained without the perspective loss.
- $S_5$: The model trained without the consistency loss.
- $S_6$: The model trained without the perspective branch, perspective loss, or consistency loss.

Table 2 shows the mAP for each variant of the proposed model. The mAP drops 3.86% without the perspective point branch ($S_1$), 1.66% without the consistency loss ($S_5$), indicating that the perspective point and re-projection consistency influence the most to the proposed framework. In addition, the switch of gate function ($S_2$, $S_3$) and perspective loss ($S_4$) contribute less to the final performance. Since $S_6$ is still higher than the state-of-the-art result [36] with 9.32%, we conjecture this performance gain may come from the one-stage (vs. two-stage) end-to-end training framework and the usage of ground-truth camera parameter; we will further investigate this in future work.

## 5.2 Heatmaps vs. Templates

As discussed in § 3.2, we test two different methods for the perspective point estimation: (i) dense prediction as heatmaps following the human pose estimation mechanism in He et al. [33] by adding a parallel heatmap prediction branch, and (ii) template-based regression by the proposed method. The qualitative results (see Figure 6) show that the heatmap-based estimation suffers severely from occlusion and topology change among the perspective points, whereas the proposed template-based regression eases the problem significantly by learning robust sparse templates, capturing consistent topological relations. We also evaluate the quantitative results by computing the average absolute distance between the ground-truth and estimated perspective points. The heatmap-based method has a 10.25 pix error, while the proposed method only has a 6.37 pix error, which further demonstrates the efficacy of the proposed template-based perspective point estimation.

Table 1: Comparisons of 3D object detection on SUN RGB-D (AP).

|  | bed | chair | sofa | table | desk | toilet | bin | sink | shelf | lamp | mAP |
|---|---|---|---|---|---|---|---|---|---|---|---|
| 3DGP [49] | 5.62 | 2.31 | 3.24 | 1.23 | - | - | - | - | - | - | - |
| HoPR [38] | 58.29 | 13.56 | 28.37 | 12.12 | 4.79 | 16.50 | 0.63 | 2.18 | 1.29 | 2.41 | 14.01 |
| CooP [36] | 63.58 | 17.12 | 41.22 | 26.21 | 9.55 | 58.55 | 10.19 | 5.34 | 3.01 | 1.75 | 23.65 |
| Ours (w/o. cam) | 71.39 | 34.94 | 55.63 | 34.10 | 14.23 | 73.73 | 17.47 | 34.41 | 4.21 | 9.54 | 34.96 |
| Ours (full) | **79.69** | **40.42** | **62.35** | **44.12** | **20.19** | **81.22** | **22.42** | **41.35** | **8.29** | **13.14** | **39.09** |

Table 2: Ablative analysis of the proposed model on SUN RGB-D. We evaluate the mAP for 3D object detection.

| Setting | $S_1$ | $S_2$ | $S_3$ | $S_4$ | $S_5$ | $S_6$ | Full |
|---|---|---|---|---|---|---|---|
| mAP | 35.23 | 38.63 | 38.87 | 39.01 | 37.43 | 32.97 | **39.09** |

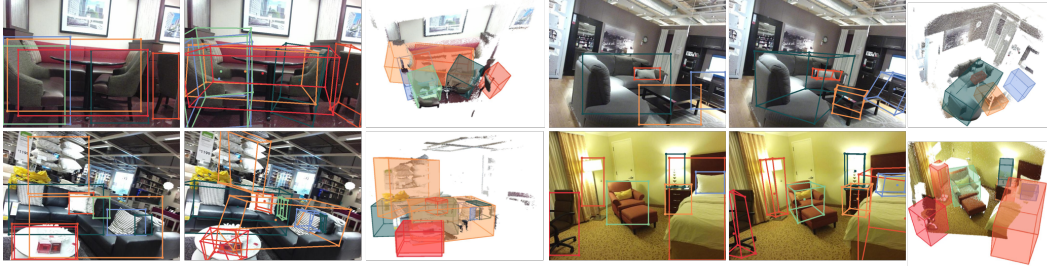

Figure 7: Some failure cases. The perspective point estimation and the 3D box estimation fail at the same time.

### 5.3 Failure Cases

In a large portion of the failure cases, the perspective point estimation and the 3D box estimation fail at the same time; see Figure 7. It implies that the perspective point estimation and the 3D box estimation are highly coupled, which supports the assumptions that the perspective points encode richer 3D information, and the 3D branch learns meaningful knowledge from the 2D branch. In future work, we may need a more sophisticated and general 3D prior to infer the 3D locations of objects for such challenging cases.

### 5.4 Discussions and Future Work

**Comparison with optimization-based methods.** Assume the estimated 3D size or distance is given, it is possible to compute the 3D bounding box with an optimization-based method like efficient PnP. However, the optimization-based methods are sensitive to the accuracy of the given known variables. It is more suitable for tasks with smaller solution spaces (*e.g.*, 6-DoF pose estimation where the 3D shapes of objects are fixed). However, it would be difficult for tasks with larger solution spaces (*e.g.*, 3D object detection where the 3D size, distance, and object pose could vary significantly). Therefore, we argue that directly estimating each variable with constraints imposed among them is a more natural and more straightforward solution.

**Potential incorporation with depth information.** The PerspectiveNet estimates the distance between the 3D object center and camera center based on the color image only (pure RGB without any depth information). If the depth information was also provided, the proposed method should be able to make a much more accurate distance prediction.

**Potential application to outdoor environment.** It would be interesting to see how the proposed method would perform on outdoor 3D object detection datasets like KITTI [84]. The differences between the indoor and outdoor datasets for the task of 3D object detection lie in various aspects, including the diversity of object categories, the variety of object dimension, the severeness of the occlusion, the range of the camera angles, and the range of the distance (depth). We hope to adopt the PerspectiveNet in future to the outdoor scenarios.

## 6 Conclusion

We propose the PerspectiveNet, an end-to-end differentiable framework for 3D object detection from a single RGB image. It uses perspective points as an intermediate representation between 2D input and 3D estimations. The PerspectiveNet adopts an R-CNN structure, where the region-wise branches predict 2D boxes, perspective points, and 3D boxes. Instead of using a direct regression of 2D-3D relations, we further propose a template-based regression for estimating the perspective points, which enforces a better consistency between the predicted 3D boxes and the 2D image input. The experiments show that the proposed method significantly improves existing RGB-based methods.

**Acknowledgments** This work reported herein is supported by MURI ONR N00014-16-1-2007, DARPA XAI N66001-17-2-4029, ONR N00014-19-1-2153, and an NVIDIA GPU donation grant.

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
