[Supplementary Material]

# *Supplementary Material* for
# 3D Object Detection from a Single RGB Image via Perspective Points

## 1   3D Object Detection

Here, we report the 3D object detection results of all 30 object categories in Table 1.

Table 1: 3D object detection on SUN RGB-D.

| toilet | recycle_bin | night_stand | endtable | drawer | computer | keyboard | table | chair | monitor | stool |
|--------|-------------|-------------|----------|--------|----------|----------|-------|-------|---------|-------|
| 81.22 | 37.68 | 35.16 | 19.77 | 1.28 | 1.24 | 2.86 | 44.12 | 40.42 | 1.14 | 22.65 |
| lamp | dresser | picture | garbage_bin | shelf | sofa_chair | cabinet | sink | desk | bookshelf | coffee_table |
| 13.14 | 27.38 | 0 | 22.42 | 0.97 | 51.86 | 1.70 | 41.35 | 20.19 | 8.29 | 28.80 |
| box | sofa | whiteboard | bed | pillow | paper | painting | cpu | mAP | | |
| 1.64 | 62.35 | 0.02 | 79.69 | 11.36 | 0 | 0.17 | 21.60 | 22.69 | | |