[Reviews · NeurIPS 2019]

Reviewer 1



- The novelty of this work lies in the insight that perspective points could serve as a better geometric constraint for 3d object detection when used as an intermediate representation as opposed to 2d bounding boxes which were used in prior work. Although such a representation has been utilized in applications such as pose estimation, it is still interesting to see it being applied for 3d detection. - Also, in this work, the projective points are inferred as a mixture of sparse templates instead of dense heatmap-based predictions for each point independently. It has been shown to be more effective with an ablation study. The perspective loss also was defined under the assumption that each 3d bounding box aligns with a local Manhattan frame. These choices made in system development has been presented well with clear writing, related work and supported with experiments.

Reviewer 2



The template-based prediction is novel, the writing is clear and the overall performance is significant. Other detailed comments: (1) It's better to clarify the design details of the perspective templates. Are templates in the same class have different pose? If so, what are the poses? (2) Sec. 5.2 heatmaps vs templates. The authors compares with the keypoint prediction baseline of Mask-RCNN, which uses the same backbone as the proposed method. I'm wondering if the worse performance of heat map based approach is due to the backbone network design. E.g., use hourglass network [A] (which is commonly used for heat map prediction) instead and see the performance? [A] Newell, Alejandro, Kaiyu Yang, and Jia Deng. "Stacked hourglass networks for human pose estimation." European conference on computer vision. Springer, Cham, 2016.

Reviewer 3



Originality: To the best of my knowledge, using projected 3D bounding box corners as an intermediate representation is a novel idea. Moreover, this is much more intuitive and natural compared to previous works. The related works are very well cited, making this paper more informative. Quality: The paper is technically sound. By introducing projected perspective points, this work achieves state of art 3D detection result on a challenging dataset. However, several ambiguities arise in the experiment section, which makes some important details less clear. 1. How are the templates defined? It seems like the templates are defined per-class and through out the paper there seems less description on how such templates are derived. Are they hand crafted or derived from the statistics of the dataset? Or are they actually optimizable so that they are actually learnt along the training? 2. Though it is nice to introduce intermediate representations, but it seems less intuitive if such representation is class conditioned. According to David Marr, as cited in the paper, such intermediate representation should be coming from lower level signals to form higher level representation or reasoning. Such as edges and depths comes from images and then they form the notion of objects or class. By adding higher level constraint (a hard constraint as the templates are class conditioned) in such representation seems less aligned with Marr's theory. Therefore, the question is what would happen if this representation is class agnostic? 3. Is 3D bbox branch really necessary? If the perspective points are given, then it is possible to directly have a MSE estimate of the 3D bbox that minimize the projection error. For example, as the 3D projection is template based, the correspondence between projection and a 3D cube is automatically given. Then, it is possible to formulate the problem into perspective n point and solve the linear system via efficient pnp. Or, one can directly represent the variables as rotation angles, distance and scales as unknown variables and solve a non-linear sum of squares problem to give an estimate of 3D bbox. Therefore it seems less intuitive why there's need to introduce the 3D bbox branch in the first place. Clarity: The paper is well written and easy to read. Though it would be better if it is more clear about some details mentioned above. Significance: I think predicting 3D properties by their projections is the right way to go and is the direction that the 3D vision community needs to hear more about. I believe future researchers should be more comfortable with this concept and use this as their default setup. Clarity: Significance:

[Author Response · NeurIPS 2019]

We'd like to express our gratitude towards *all* the reviewers who recognized the novelties of the proposed intermediate
representation for 3D object detection and the template-based prediction, as well as the significantly improved
performance. We further appreciate R3 for commenting that *"predicting 3D properties by their projections is the right*
*way to go and is the direction that the 3D vision community needs to hear more about. I believe future researchers*
*should be more comfortable with this concept and use this as their default setup."*

**R1, R2, R3: More design details about the templates. Are templates in the same class have different poses?**
The templates design is both class-specific and instance-specific: (1) Class-specific: we decouple the prediction of
the perspective point and object class as illustrated in Eq.2 and L158-L159. (2) Instance-specific: the templates
$T \in \mathbb{R}^{C \times K \times 2 \times 9}$ are inferred for each RoI; hence, they are specific to each object instance as shown in Fig.3 and
L152-L153. The templates are automatically learned for each object instance from data with the end-to-end learning
framework; thus, both the templates and coefficients for each instance are optimizable and can fit the data better. We
will make the description of the class-specific and instance-specific design more clear in revision using the extra page.

**R3: What would happen if the intermediate representation is class agnostic?** Very insightful observation. We
theoretically and empirically explain our class-specific and instance-specific design. **Theoretically**, if we want to stay
with Marr's theory [1], the 3D shape should be represented by class-agnostic *3D primitives* and class-specific *3D model*
*descriptions* of a shape. By analogy, they are similar to the templates of perspective points and their coefficients in
this paper, respectively. Hence, the intermediate representation should be class-agnostic by Marr's theory. However,
the class-agnostic design would encourage the competition across classes during training, and the data we use in SUN
RGB-D dataset is imbalanced (rare objects in certain categories). To avoid such competition, we make the intermediate
representation **class-specific**. Similar with defining general 3D primitives like cuboid and cylinder, if we could design
certain class-agnostic templates by analyzing the manifold in the projected 2D space (without data-driven method), it
would be an exciting direction to see if it could learn the intermediate representation with such a class-agnostic way.
Meanwhile, in complex indoor scenes with man-made objects, the number of object categories is large (>30 categories)
with significant intraclass variations of appearance and geometry (*e.g.*, desk, lamp, and cabinet). Given a limited (5k)
and imbalanced training data, it would be challenging to model the objects with rare appearance, 3D dimensions, and
shapes if we purely use shared-templates within the same class. Therefore, we adopt an **instance-specific** template
design to model the complex data distribution. **Empirically**, the results from the class-agnostic design tend to fit the
distribution of most frequent object categories, making the performance of the rare objects much worse, and the pix
error of the perspective point estimation much larger. Our experiments also indicate that an instance-specific design
improves performance. However, a more principled method to define the template would be a combination of the
shared-templates and instance-specific templates (similar with [70]), which would be a promising future direction. We
will clarify and discuss the template design in revision using the extra page.

**R3: Is 3D bounding box branch necessary? Comparison with optimization-based prediction.** Two motivations:
(1) The **single view ambiguity** due to projection. Given only the results from 2D box branch and perspective points
branch, it is almost impossible to get a unique solution of the 3D bounding boxes; there exist multiple 3D bounding
boxes with different sizes and distances that could be projected to the very same perspective points. (2) Assume the
estimated 3D size or distance is given, it is possible to compute the 3D bounding box with an optimization-based
method like efficient PnP. However, the optimization-based methods are sensitive to the accuracy of the given known
variables. It is more feasible for tasks with smaller solution spaces (*e.g.*, 6-DoF pose estimation where the 3D shapes of
objects are fixed), But it would be difficult for tasks with larger solution spaces (*e.g.*, 3D object detection where the
3D size, distance, and object pose could vary greatly). Therefore, we argue that directly estimating each variable with
constraints imposed among them is an easier and more straightforward solution.

**R1: Applying the proposed method to an outdoor environment will be interesting.** We concur. It is interesting
to see how our method will perform on outdoor 3D object detection dataset like KITTI. The differences between the
indoor and outdoor dataset for the task of 3D object detection lie in various aspects including the diversity of object
categories, the variety of object dimension, the severeness of the occlusion, the range of the camera angles, and the
range of the distance (depth). We hope to adopt the proposed method in the future to the outdoor with ablation studies.

**R1: Potential incorporation with depth information will be interesting.** The proposed method estimates the
distance between the 3D object center and camera center based on visual features (RGB without depth) only. If the
depth were also provided, the proposed method would be able to make a much more accurate distance prediction.
In theory, the depth information would help to improve the overall performance significantly, but it would make the
problem less challenging or interesting, in our opinion. However, we hope to explore a better way to incorporate depths.

**R2: Backbone design for the heatmap-based approaches may influence the perspective point estimation.** It is
possible to devise a two-stage method—detect the object in the first stage and infer the perspective point in the second
stage with [34]. However, [34] is not originally designed for regressing the keypoints for multiple categories of objects.
We will try our best to include an additional ablative study by comparing with networks similar to [34] in revision.

[Meta-Review · NeurIPS 2019]

Three reviewers recommend acceptance. The approach is sound and obtains good results. Good job.